# Dynamics in Quality of Life of Breast Cancer Patients Following Breast-Conserving Surgery Versus Mastectomy: Protocol for Systematic Review and Meta-Analysis

**DOI:** 10.3390/healthcare12222288

**Published:** 2024-11-15

**Authors:** Iryna Makhnevych, Darya Smetanina, Mohamed Fatihy Elgasim Abdelgyoum, Jood Jasem Shaddad Alblooshi, Aysha Khamis Alhosani, Ibrahim Mohamed Ahmed Musa, Anna Nimer, Kornelia Zaręba, Tallal Younis, Milos Ljubisavljevic, Yauhen Statsenko

**Affiliations:** 1Department of Radiology, College of Medicine and Health Sciences, United Arab Emirates University, Al Ain P.O. Box 15551, United Arab Emirates; iramakhnevych@gmail.com (I.M.); daryasm@uaeu.ac.ae (D.S.); fatihymhmd288@gmail.com (M.F.E.A.); joodalblooshi@gmail.com (J.J.S.A.); aishaalhosani21@gmail.com (A.K.A.); hema2030hema@gmail.com (I.M.A.M.); 2Medical Imaging Platform, ASPIRE Precision Medicine Institute in Abu Dhabi, Imaging Platform, United Arab Emirates University, Al Ain P.O. Box 15551, United Arab Emirates; 3Department of Obstetrics and Gynecology, College of Medicine and Health Sciences, United Arab Emirates University, Al Ain P.O. Box 15551, United Arab Emirates; animer@uaeu.ac.ae (A.N.); kornelia3@poczta.onet.pl (K.Z.); 4Department of Internal Medicine, College of Medicine and Health Sciences, United Arab Emirates University, Al Ain P.O. Box 15551, United Arab Emirates; tallal.younis@uaeu.ac.ae; 5Department of Physiology, College of Medicine and Health Sciences, United Arab Emirates University, Al Ain P.O. Box 15551, United Arab Emirates; milos@uaeu.ac.ae; 6Neuroscience Platform, ASPIRE Precision Medicine Institute in Abu Dhabi, Imaging Platform, United Arab Emirates University, Al Ain P.O. Box 15551, United Arab Emirates

**Keywords:** breast cancer, quality of life, QoL dynamics, long-term QoL, breast-conserving surgery, breast reconstruction, mastectomy, meta-regression

## Abstract

**Background:** Quality of life (QoL) may change in female patients with breast cancer over time due to its dynamic construct. Recent studies have provided statistics on the diverse predictors of QoL in breast cancer patients. Still, the literature findings on the prognostication of QoL are scarce. **Objectives:** With this meta-analysis, we aim to describe the dynamics of overall QoL and its dimensions: mental status, physical functioning, and social interactions. **Methods:** To reach this aim, we will systematically evaluate the peer-reviewed literature on QoL of women who have undergone either mastectomy or breast-conserving surgery. The proposed study will focus on, but not be limited to, the analysis of BIRS, FACT-B, and EORTC QLQ-C30 questionnaires, which are most commonly used in the assessment of the quality of life of cancer patients. Then, we will extract the following outcome measures: The participants’ age, time since surgery, type of tumor removal procedure (mastectomy or breast-conserving surgery), breast reconstruction technique, mean value, and standard deviation in a QoL score. A random-effects model will then compute the pooled QoL and construct the trend lines for scores received from each diagnostic tool. The findings will be adjusted according to the reconstruction techniques and tumor removal surgery. Finally, we will model the QoL dynamics with a set of predictors identified in the extracted studies. **Conclusions:** The study findings may serve as a tool for stratifying female patients with breast cancer by the risk of significant reduction in QoL.

## 1. Introduction

Breast cancer is the most common malignancy in women worldwide [1]. The death rate for the disease is higher in transitioning than in transitioned countries [2]. In 2017, the total macroeconomic cost of breast cancer was estimated at USD 2.0 trillion [3], and it is expected to reach USD 25.2 trillion by 2050 [3,4]. In developed countries, the mortality rate for breast cancer dropped by 40% in the last two decades of the 20th century. Still, the cancer burden exerts tremendous physical, emotional, and financial strain on individuals, families, and communities [5,6].

Oncologists should consider patient risks of developing mood disorders and adjust their surgical approaches accordingly. However, physicians fail to reliably prognosticate psychological responses to tumor removal and breast reconstruction. The following solutions can be adopted to improve the current situation. First, the quality of life (QoL) assessment should cover multiple domains since cancer treatment may affect single or multiple dimensions of well-being [7]. Second, the models prognosticating QoL should reflect the impact of different contributing factors. These factors comprise case history, demographics, the time lapse from the treatment, and accessibility of informational, emotional, and tangible support to the patient [8,9,10,11]. Modeling QoL dynamics after surgery for breast cancer is the issue of ongoing studies [12,13,14].

The diversity of tools for QoL assessment complicates the comparison of individual studies [15]. Commonly, clinical psychologists use the following three questionnaires: The Body Image and Relationship Scale (BIRS), Functional Assessment of Cancer Therapy-Breast (FACT-B), and the European Organization for Research and Treatment of Cancer Quality of Life Questionnaire (EORTC QLQ-C30). These tools differ in the domains they assess. The BIRS measures total QoL, social barriers, strength and health, appearance, and sexuality [16]. The FACT-B is a comprehensive tool for evaluating physical, social/familial, emotional, and functional well-being. It contains a breast cancer subscale that measures QoL of patients with this specific diagnosis [17]. The aforementioned questionnaires do not reflect the type of tumor removal surgery and breast reconstruction technique, which limits the prognostic value of these assessment tools.

The type of surgery and its indications impact QoL after breast removal. Breast reconstruction can improve the patient’s life, and the results are better after the application of an autologous than an alloplastic technique [18,19,20]. Nevertheless, the optimal time and material for the reconstruction remain undiscovered. To improve the appearance of the breast, surgeons resort to minimally invasive techniques such as skin- and nipple-sparing mastectomy, acellular dermal matrices, and prepectoral breast reconstruction [21]. Oncoplastic breast-conserving surgery (OBCS) is an option for patients requiring breast surgery: it applies plastic surgery principles, thus minimizing the trauma and preventing future local deformities [22]. Some surgical techniques have recently been introduced, but any reliable statistics on QoL after their implementation is still missing.

The findings of recent studies on QoL prognosis cannot be easily applied in practice due to the diverse research methodology and duration of observations. A detailed meta-analysis helps to address several limitations of cross-sectional studies and combine available findings in a single comprehensive report. Pooling data from multiple studies helps to construct regression slopes depicting trends in QoL after surgery [23]. The trend lines can be constructed for various QoL domains assessed with distinct tools in patients who have undergone either mastectomy or breast-conserving surgery. A graphical presentation of QoL trends simplifies data interpretation and facilitates decision-making. This serves as a motivation for the current research. We will examine the relationship between the time since surgery and patients’ quality of life using a meta-regression approach.

## 2. Objective

The aim of the planned systematic review and meta-analysis is to explore the impact of the time after mastectomy or breast-conserving surgery on different aspects of the patient’s well-being. The study’s working hypothesis is that a thorough analysis of biological, demographic, psychosocial, and clinical predictors will allow us to prognosticate the dynamics of different QoL estimates after surgery. Once created and tested, a meta-regression model will serve as a tool for stratifying female patients with breast cancer by the risk of a dramatic reduction in QoL.

To achieve the study aim, we formulated the following specific objectives:Model the dynamics of QoL estimates in breast cancer patients after surgery.Perform a subgroup analysis to assess the impact of breast-conserving surgery and mastectomy with different breast reconstruction techniques on QoL.Explore the effect of other confounding factors on QoL after breast tumor removal. The study will cover clinical, demographic, and socioeconomic confounders.

## 3. Materials and Methods

The protocol is prepared according to the checklist of the Reporting Items for Systematic Review and Meta-analysis Protocol (PRISMA-P) [24], which is available in Appendix A. The study is filed with the international database for systematic reviews, PROSPERO, with the registration number CRD42024565182. Any changes required during the review preparation process will be reported by updating the online registered PROSPERO protocol.

### 3.1. Study Design and Data Source

We will conduct a comprehensive literature search in the biomedical databases Scopus, CINAHL, Embase, APA PsycArticles, PubMed, SciELO, LILACS, and Global Index Medicus. The keywords and medical subject headings are listed in Table 1.

### 3.2. Eligibility Criteria

The systematic review will examine QoL in female patients with breast cancer after curative-intent surgical treatment: mastectomy vs. breast-conserving surgery (BCS). Specifically, we will focus on cancer survivors, i.e., the patients who have completed initial cancer treatment, whose disease is either non-active or progressive but in a non-terminal stage [25]. The final review will analyze original peer-reviewed publications, theses/dissertations, official reports of healthcare associations, and other materials relevant to the study objective. We will exclude protocol papers, communications, editorial letters, and conference abstracts. Although the conference presentations may contain relevant findings, they can miss confounding factors and crucial details on methodology. To obtain full information on these studies, we will manually search for full-length articles reporting the data. We will consider papers written in any language and published between 2000 and 2024.

The criteria for including the articles in the review are as follows. The articles should report scores in any type of questionnaire for QoL assessment. The study cohorts should consist of participants free from chronic mental disorders known before the cancer diagnosis. The participants should not be victims of domestic violence or any other abuse before the cancer diagnosis, during treatment, and in remission. We will exclude papers reporting COVID-19-related aspects of QoL and publications about the role of psychological interventions on QoL. We will not limit the search to a particular follow-up period; the study aims to gather information about QoL at different time points. This approach will allow us to depict fluctuations in QoL over the lifespan.

### 3.3. Selection Process

The online systematic review software Covidence will be used to manage the selection of eligible articles [26]. Publications retrieved from the databases will be uploaded to the software for automatic deduplication. Two reviewers will screen titles and abstracts against the inclusion and exclusion criteria. If the reviewers disagree on the eligibility of a study, the issue will be resolved in a discussion between them and the principal investigator. The same procedure will be applied to the full-text screening. The reasons for excluding articles will be recorded and demonstrated in a PRISMA flowchart.

### 3.4. Data Extraction

From each study, two authors will independently extract data in a tailored template. The information will cover study characteristics, methodology details, and findings. Basic characteristics will encompass the author’s name, year of publication, the country where the study was conducted, and potential conflicts of interest. The methodology details will report the study design, period of data collection, study aim/objectives, inclusion and exclusion criteria for the participants, assessment tools, and sample size. The study results should report the participants’ age, time since surgery, type of tumor removal procedure (mastectomy or breast-conserving surgery), breast reconstruction technique, mean value, and standard deviation in a QoL score. We will also extract the available information about breast reconstruction techniques and other confounders that can affect study results. Table 2 presents the full list of target variables.

### 3.5. Quality Assessment of Studies

Two authors will independently perform the quality appraisal of the publications with Study Quality Assessment Tools developed by the National Institute of Health [27]. If any disagreements arise, the principal investigator will make the final decision regarding the publication’s quality score.

Publication bias will be assessed with graphical and statistical methods. Begg’s and Egger’s tests will be used to construct funnel plots for a visual presentation of reporting bias [28,29]. The asymmetry of the plots indicates between-study heterogeneity. The “trim and fill” method will identify the number of studies required to construct a symmetric funnel plot [30].

The studies that will be included in the review may differ in quality and statistical approaches to designing and addressing a research question. To ensure the robustness of the meta-analysis, we will perform the following actions. First, we will assess the weighted contributions of individual publications to minimize the impact of medium-quality studies on the overall analysis. Second, a sensitivity analysis will be performed to check whether the medium-quality publications or studies with mixed results may have affected the final results. Third, we will report the effect of the above-mentioned publications and studies on our findings.

### 3.6. Inter-Rater Reliability

Different techniques for calculating inter-rater reliability will be applied throughout the meta-analysis. After screening the titles/abstracts and the full texts, we will use Covidence to export the inter-rater reliability report. It contains proportionate agreement between the reviewers, probability of “YES” and “NO” votes, random agreement probability, and Cohen’s Kappa coefficient [31]. A coefficient value greater than 70 will indicate that the criteria are clear and objective and that their application to the selection process, data extraction, and quality appraisal is consistent [32]. Since the results of data extraction and quality assessment will not be recorded in Covidence, we will use R package “irr” to calculate the inter-rater reliability for these stages [33,34].

### 3.7. Data Analysis and Synthesis

Before the data analysis, we will check the extracted data for homogeneity via Cochrane’s chi-squared and $I^{2}$ statistic. $I^{2}$ values of 50–75% and ≥75% would signal substantial and considerable heterogeneity, respectively [35,36]. To determine the moderator’s role in heterogeneity, we will use a mixed-effects meta-regression approach [37]. The following variables may constitute the principal sources of heterogeneity: The age of the participants, sample size, time passed since the surgery, type of breast surgery, and reconstruction techniques. Leave-one-out cross-validation will be used to verify the robustness of the overall study results and the influence of each included study on the outcomes of the meta-analysis.

As part of working on the first specified objective, we will use a random-effects model to compute the pooled QoL. The pooled value will estimate variance across the results in individual studies at each time point. Then, we will describe the long-term QoL dynamics and separately construct the trend lines for any findings received from different QoL assessment tools.

To model the time evolution of QoL, we will consider linear, quadratic, cubic, or higher-degree equations [see Equations (Equation 1)–(Equation 4)]. An alternative way would be to use hybrid models with exponential cumulative distributions for growth with the linear, quadratic, cubic, or higher-degree equations [see Equations (Equation 5)–(Equation 7)]. Then, we will select the model explaining most of the data with a minimum number of parameters. To identify the best one among the candidate models, we will use a Bayesian information criterion.
(1)$$QoL\_estimate=\beta_{0}+\beta_{1}Time+\epsilon$$
(2)$$QoL\_estimate=\beta_{0}+\beta_{1}Time+\beta_{2}Time^{2}+\epsilon$$
(3)$$QoL\_estimate=\beta_{0}+\beta_{1}Time+\beta_{2}Time^{2}+\beta_{3}Time^{3}+\epsilon$$
(4)$$QoL\_estimate=\beta_{0}+\beta_{1}Time+\beta_{2}Time^{2}+\ldots +\beta_{k}Time^{k},k=1,10.$$
(5)$$QoL\_estimate=\beta_{4}\left(1-e^{-Time/\beta_{5}}\right)+\beta_{0}+\beta_{1}Time+\epsilon$$
(6)$$QoL\_estimate=\beta_{4}\left(1-e^{-Time/\beta_{5}}\right)+\beta_{0}+\beta_{1}Time+\beta_{2}Time^{2}+\epsilon$$
(7)$$QoL\_estimate=\beta_{4}\left(1-e^{-Time/\beta_{5}}\right)+\beta_{0}+\beta_{1}Time+\beta_{2}Time^{2}+\beta_{3}Time^{3}+\epsilon$$

To address the second specified objective, we will divide the findings according to the reconstruction techniques performed after mastectomy or BCS. Then, we will analyze each subgroup in the same manner as the total sample.

To complete the third specified objective, we will model the QoL dynamics. The research team will extract all the determinants from the literature and consider them as predictors (see Table 2). If the percentage of missing values is less than 30%, we will apply an imputation technique to generate the values. Then, our team will normalize numerical variables by subtracting the mean value and scaling the result to the attribute variance. With feature selection techniques, we will select the top informative predictors to optimize model performance. It will be assessed with $R^{2}$, root-mean-squared deviation, and the mean absolute error divided by the range of values (MAE/ROV). A regression model will be used to eliminate the effects of confounding factors on study results. For the statistical analysis, we will resort to R package “meta” and python packages [38].

## 4. Discussion

### 4.1. Dynamics of QoL in Breast Cancer Patients After Surgery

The duration of the follow-up period impacts patient’s QoL. In one case study, patients completed the EORTC QLQ-C30 questionnaire three months after the surgery. The study showed a gradual improvement in the global health status and physical, emotional, social, and role functioning [39]. In the interval from one to five years, another study reported improved physical function, body image, and sexuality as per the CARES questionnaire [40]. Still, the comparison of individual findings is challenging because of the different assessment tools used to examine life quality.

Recent studies advocate for BCS. Some of them showed that the aesthetic outcome is better after BCS than complete breast removal [41]. Three months after the surgery, BCS patients scored higher in a WHO QoL BREF questionnaire than those who underwent modified radical mastectomy [42]. However, another study showed the opposite findings in the late follow-up period. In a five-year-long perspective, the global health status was higher after mastectomy than BCS. Still, BCS patients scored higher in some scales of EORTC QLQ-C30 [43]. Combined with radiotherapy, BCS showed a better 10-year relative survival rate compared to mastectomy [44]. Despite the advantage of conventional BCS over mastectomy, it has some contraindications: local metastasis, diffuse microcalcifications, irradiated thoracic wall, the first two trimesters of pregnancy, and mutations in *BR-CA1* and *BR-CA2* genes [41]. This limits the applicability of the interventions that remove breast cancer while avoiding mastectomy.

### 4.2. Clinical Determinants of QoL After Intervention

The major clinical determinants of QoL are cancer type, stage, breast surgery, and adjuvant therapy, which may include chemotherapy, radiation therapy, and hormone therapy. Following mastectomy, women can undergo autologous or alloplastic breast reconstruction. The first technique allows for the transplantation of a personal flap from the abdomen, back, inner thigh, or buttocks to the breast. Silicone or saline implants are the options for alloplastic surgery. The QoL outcomes are better after autologous than alloplastic reconstruction. A recent meta-analysis supported this fact by reporting major complications and reconstructive failure in 60–80% of women with alloplastic reconstruction. These conditions required surgical correction and negatively impacted QoL [45].

OBCS is a new trend in BCS that merges oncology and plastic surgery to achieve both therapeutic and aesthetically satisfying outcomes. During OBCS, surgeons resort to the techniques of volume displacement or replacement. The latter requires the personal tissue that can be obtained from different loci: local perforator flaps, latissimus dorsi muscle, free flaps, or areas with sufficient amount of fat [46]. The results may differ across the types of autotransplant in post-surgical complications, visual outcomes, and women’s perception about their body image. Information on the QoL outcomes of OBCS is still limited: a former study showed high scores in BREAST-Q after volume displacement surgery [47]. We failed to find a systematic review that compares QoL after volume displacement and replacement techniques.

The type and timing of reconstruction surgery are potential confounders of QoL. A systematic review dealt with well-being after delayed and immediate breast reconstruction. The authors reported a non-pronounced difference in QoL outcomes of these interventions [48]. Despite the advantage of immediate breast reconstruction, it can be contraindicated, especially after radiotherapy [49]. For example, in alloplastic surgery, radiation may cause implant loss, wound-healing problems, and reparations. Meanwhile, in autologous breast reconstruction, radiation may lead to flap shrinkage, necrosis, and unfavorable aesthetic results [50]. The impact of reconstruction surgery timing on QoL remains unclear and it should be studied in future systematic reviews.

Another potential QoL confounder is the patient’s involvement in the decision on the reconstruction technique. Patients’ engagement in decision-making is associated with better QoL, and vice versa, low patient involvement may lead to decision regret [48]. Other risk factors of remorse include low socioeconomic status, improper physician–patient communication, lack of information on potential complications, and inadequate exploration of patient expectations [51]. The scientific community has not built a consensus about optimal patient management that would reduce the risk of decision regret in breast cancer patients. Recent studies on this issue have a common shortcoming: clinicians do not document the conditions in which the decision on the reconstruction technique is carried out [51]. This limitation may hinder the actual reason behind the QoL dynamics.

Adjuvant therapy may also impact QoL. The therapy is given in addition to the primary treatment to maximize its effectiveness. Some patients undergo surgery, chemo-, and hormonal therapy, and they have better QoL outcomes than those who receive radiation therapy [52,53]. The association of QoL dynamics with treatment modalities is scarcely presented in the available literature.

### 4.3. Other Confounders of QoL After Breast Tumor Removal

The determinants of QoL in breast cancer patients are diverse, and they fall into several categories: clinical determinants, demographic, socioeconomic, and psychological risks (see Table 2). Findings on the role of age in QoL changes are inconclusive among different studies. Some of them indicated that women below 45 years showed poorer QoL [54,55,56,57,58,59,60]. Others concluded that QoL was better in younger patients compared to women of older ages [52,61]. The impact of demographics on QoL should be investigated in the proposed systematic review.

The importance of the socioeconomic predictors of QoL has been justified in previous studies. For example, QoL is better in women with higher education, income, and active employment [62,63,64]. Marital status also plays an important role in QoL; single, divorced, and widowed women have poorer QoL compared to married women [55,62]. Being a minority may also impact QoL [65]. Data on socioeconomic status must be used as predictors of QoL after breast cancer treatment.

Psychological determinants of QoL comprise personality traits, psychological interventions, emotional support, etc. Personality traits may play a crucial role in the QoL outcomes of breast surgery. For instance, better QoL is linked with optimism, sociability, affability, active coping strategies [66,67], sense of coherence, and self-efficacy [62,68,69,70]. Contrarily, poorer well-being is associated with lower emotional intelligence [71] and mental disorders [72].

Many studies emphasize the importance of psychological support for breast cancer patients [73]. Social isolation and loneliness have a detrimental impact on QoL [74,75]. Psychological nursing can significantly reduce the patient’s fear and enhance their self-confidence in the face of cancer [76]. Cognitive behavior therapy is another feasible tool for the improvement of QoL, although it manages depression and anxiety better than enhances QoL [77]. Patient engagement in self-care activities ensures better QoL in breast cancer [78]. Socioeconomic and demographic risks may greatly modify the results of professional emotional support. Hence, all these findings should be analyzed for an accurate prognosis of QoL after intervention.

## 5. Conclusions

The increasing survival rate and incidence of breast cancer call for improved care for patients who seek the restoration of physical, psychological, and social wellness after treatment. The optimal rehabilitation program comprises the stratification of patients by the risk of negative QoL dynamics and the incorporation of psychological interventions in cancer management. Studies should assess the impact of tumor removal surgeries and breast reconstruction techniques on different aspects of Qol, including psychological and socioeconomic determinants of well-being.Currently, no study provides uniform information on the long-term dynamics of QoL in breast cancer. For this reason, we propose a meta-analysis that will discern the trends in a set of QoL estimates after breast-conserving surgery and mastectomy followed by breast reconstruction. The study findings may serve as a tool for stratifying female patients with breast cancer by the risk of significant reduction in QoL.Clinicians would appreciate a reliable tool to explore the immediate and delayed impact of treatment on various dimensions of QoL in breast cancer patients. Hypothetically, a thorough analysis of biological, demographic, psychosocial, and clinical predictors would allow us to prognosticate the dynamics of different QoL estimates after surgery.

## 6. Strength and Limitations

The strengths of the proposed study are listed below:The proposed meta-analysis will cover multiple time points since surgery. Authors will use them to identify QoL dynamics with linear, quadratic, cubic, or higher-degree equations.The study findings will enrich healthcare specialists with data on diverse tools for QoL assessment.To build a reliable meta-regression, authors will train the models on a large number of informative predictors, which will ensure a high model performance.

The weaknesses of the proposed meta-analysis are as follows:A notable limitation is the potentially high variability in methodology across original studies. This heterogeneity may limit the generalizability of our research findings to other settings and clinical practices beyond those examined in the individual studies.It is impossible to take into account the impact of cultural norms, religion, access to healthcare, and community lifestyle on QoL of women. Hence, this information on important determinants of well-being will be missing.During preliminary hand-screening, we found that some articles did not report key clinical variables affecting QoL, such as time since diagnosis and disease stage. The absence of these data is a potential source of heterogeneity.

## Figures and Tables

**Table 1 healthcare-12-02288-t001:** Keywords and medical subject headings for PubMed/Medline.

No	Search String	Number of Articles
1	((cancer survivors[MeSH Terms]) OR (cancer patients[Title/Abstract])) OR (breast cancer patients[Title/Abstract])	252,499
2	(((((breast-conserving surgery[Title/Abstract]) OR (breast conservation[Title/Abstract])) OR (breast conserving surgery[Title/Abstract])) OR (lumpectomy [Title/Abstract])) OR (lumpectomy [MeSH Terms])) OR (“mastectomy, partial”[MeSH Terms])	17,665
3	(mastectomy[MeSH Terms]) OR (mastectomy[Title/Abstract])	49,704
4	((quality of life[MeSH Terms]) OR (quality of life[Title/Abstract])) OR (well-being[Title/Abstract])	587,301
5	#1 AND (#2 OR #3) AND #4	2820 *

* Number of papers after limiting the search string to studies published between 2000 and 2024.

**Table 2 healthcare-12-02288-t002:** Determinants of quality of life in patients with breast cancer.

Group	Subgroup	Variables
Clinical determinants	Tumor-related risks	Type of breast cancer:
		- Ductal carcinoma
		- Lobular carcinoma
		- Medullary carcinoma
		- Tubular carcinoma
		- Mucinous carcinoma
		- Paget’s disease
		- Metaplastic breast cancer
		- Triple negative breast cancer
		- Inflammatory breast cancer
		- Metastatic breast cancer
		- Breast cancer during pregnancy
		- Other types of cancer
		Tumor stage and treatment:
		- Tumor stage
		- Chemotherapy
		- Radiotheraphy
	Breast removal determinants	Type of surgery:
		- Breast-conserving surgery
		- Oncoplastic breast-conserving surgery
		- Radical mastectomy
		- Skin-sparing mastectomy
		- Nipple-sparing mastectomy
		- Acellular dermal matrices
		- Prepectoral breast reconstruction
		- Other techniques
	Breast reconstruction determinants	Type of implant:
		- Auto/alloplastic
		- Implant shape
		- Implant material
		Time of reconstruction:
		- Immediate reconstruction
		- Delayed reconstruction
	Surgical risks	Complications:
		- Infection
		- Blood clot formation
		- Seroma
		- Hematomas
		- Scar tissue
		- Scar formation
		- Other complications
		- In-hospital length of stay
Other confounders	Demographic risks	Age group
		Country of study
		Race/ethnicity
	Socioeconomic risks	Socioeconomic status (low, medium, high)
		Level of education
		Income
		Country gross domestic product
		Marital status
		Employment status
	Psychological risks	Personality traits
		Psychological interventions
		Emotional support

## Data Availability

Data are contained within the article and Appendix A.

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
