# Peer review of "Dynamics in Quality of Life of Breast Cancer Patients Following Breast-Conserving Surgery Versus Mastectomy: Protocol for Systematic Review and Meta-Analysis"

_healthcare, 2024, doi:10.3390/healthcare12222288_

Round 1
Reviewer 1 Report
Comments and Suggestions for Authors
I consider the proposed search strategy to be insufficient. It would be advisable for the authors to seek assistance from a librarian specialized in biomedical sciences to improve the search strategy. There are several well-known instruments for measuring quality of life that have not been included in the search, which limits the scope of the review.
The rationale for including the term "cancer survivor" in the search strategy is unclear. It would be helpful for the authors to justify its inclusion or reconsider whether it truly fits the objective of the study.
It is suggested that the authors avoid conducting searches in PubMed using the [TEXT WORD] field, as this may yield irrelevant results. It is more advisable to restrict the search to the title and abstract fields, which will improve the precision and relevance of the studies identified.
The authors should correctly cite the programs and applications they will use in the study. For instance, they mention the use of COVIDENCE but do not cite it properly. All methodological tools used should be properly referenced.
The search should be expanded to other databases that are more specific to the field of biomedicine. For example, although Scopus is a relevant database, it is not specific to biomedical sciences. Additionally, CINAHL focuses mainly on nursing, which may not be appropriate for the central theme of the study. It is recommended to include other more relevant databases, such as Embase, and specialized ones like PsycINFO or the Cochrane Library, depending on the study's objectives.
The correct term is PubMed. If the authors decide to mention PubMed/Medline, they should include a specific subset, as these terms are not synonymous and can cause confusion.
Errors were detected in the checklist presented by the authors. For example, they mention that they will not conduct sensitivity analysis, yet the manuscript discusses "leave-one-out cross-validation," which is a common sensitivity analysis technique. These inconsistencies need to be corrected, and the checklist should be adjusted to reflect the characteristics of the proposed study.
The proposed study does not adequately address its limitations. The authors should consider additional limitations, such as those inherent in the use of certain databases, which can influence the thoroughness and quality of the results. Furthermore, it would be important to discuss the limitations related to the search methodology, study selection, and potential heterogeneity.
The authors do not mention the reason for excluding gray literature from their review. It is essential to justify this decision, as gray literature can be an important source of information for certain topics, especially in emerging research areas.
Finally, it is suggested that the authors include an explicit statement that any changes to the protocol or search strategy will be registered and updated in PROSPERO to ensure transparency in the systematic review process.
Comments on the Quality of English LanguageMinor editing of English language required.
Author Response
Thank you for the review. Kindly find attached "Response_Reviewer_1.docx" file.

Reviewer 2 Report
Comments and Suggestions for Authors
suggested revisions points for the provided protocol
- Strengthen the rationale for the study by connecting the identified gaps to the proposed study more clearer.
- Add more discussion on the clinical implications of your analysis findings or how the proposed protocol covers all aspects of the subject
- Include a more detailed explanation of the meta-regression model, including the variables that will you handle
- List the steps for ensuring inter-rater reliability between reviewers during the study selection and data extraction phases.
- Discuss the details methods to detect or avoid any publication bias.
Author Response
Thank you for the review. Kindly find attached "Response_Reviewer_2.docx" file.

Reviewer 3 Report
Comments and Suggestions for Authors
Dear authors,
Your manuscript looks more like a study design proposal. I see no purpose in publishing this. My recommendations would be to undergo the research and publish a manuscript with the results.
Author Response
Dear reviewer,
We appreciate your efforts made to review the manuscript. It is a protocol article announcing upcoming systematic review, discussing study methodology, strength and limitations with the field experts. Prior to the submission, we wondered if MDPI publishes protocol articles, and the official reply was positive. Once reviewed and published, the protocol paper serves as a valuable planning and quality assurance tool providing more structure to the review process.
Kindly see the official information at https://www.mdpi.com/about/announcements/2222 . It literally tells, that Study Protocol is a new article type accepted for submission:
"
We are pleased to announce that we are starting to accept submissions of Study Protocols, to be considered for publication.
Study Protocol manuscripts report proposed or ongoing prospective research. The publication of study protocols can reduce publication bias and improve the reproducibility of research. This also helps to prevent the unnecessary duplication of effort and work. Preference will be given to submissions describing long-term studies and those likely to generate a considerable amount of outcome data. If data collection is complete, the manuscript will not be considered. Study protocols for pilot or feasibility studies, or if the authors have other articles relating to the protocol published or under consideration, are also not considered. Authors are encouraged to later publish the study results as research articles, in any journal of MDPI. The research articles will receive a 20% discount on the APCs (type: article, 2 max.). The research article and the original study procotol will then be linked.
Study Protocols should use the template file to prepare the front matter and back matter... Protocols of systematic reviews should follow the PRISMA-P guidelines."
These article should follow the Preferred Reporting Items for Systematic Reviews and Meta-analysis Protocols (PRISMA-P). Our manuscript also adheres to this guideline.
See below the list of similar protocols of systematic reviews recently published in Healthcare journal:
- Magi CE, Bambi S, Rasero L, Longobucco Y, El Aoufy K, Amato C, Vellone E, Bonaccorsi G, Lorini C, Iovino P. Health Literacy and Self-Care in Patients with Chronic Illness: A Systematic Review and Meta-Analysis Protocol. Healthcare 2024 Mar 31 (Vol. 12, No. 7, p. 762). MDPI.
- Sanchis-Soler G, Tortosa-Martinez J, Sebastia-Amat S, Chulvi-Medrano I, Cortell-Tormo JM. Is Acute Lower Back Pain Associated with Heart Rate Variability Changes? A Protocol for Systematic Reviews. Healthcare 2024 Feb 3 (Vol. 12, No. 3, p. 397). MDPI.
- de Diego-Alonso C, Blasco-Abadía J, Buesa-Estéllez A, Giner-Nicolás R, López-Royo MP, Roldán-Pérez P, Doménech-García V, Bellosta-López P, Fini N. Relationship between Participation in Daily Life Activities and Physical Activity in Stroke Survivors: A Protocol for a Systematic Review and Meta-Analysis. Healthcare 2023 Jul 30 (Vol. 11, No. 15, p. 2167). MDPI
Regards,
Yauhen Statsenko on behalf of co-authors

Reviewer 4 Report
Comments and Suggestions for Authors
This manuscript is a well-written and methodologically sound protocol for a systematic review and meta-analysis, it has significant potential to fill gaps in understanding the long-term QoL outcomes in breast cancer patients post-surgery. A few suggestions for improvements in the discussion, handling of heterogeneity, and expansion of the objectives would enhance its clarity and impact.
Title and abstract
1/ Clarify the specific time points for the quality of life (QoL) assessment and ensure the keywords are updated to better reflect the scope of the review.
Introduction
1/ More detailed discussion on recent advancements in this field, particularly new breast reconstruction techniques.
2/ Including more up-to-date references from 2021–2023 would strengthen the relevance of the study.
Methodology
1/ how discrepancies between the two independent reviewers will be handled beyond just involving the principal investigator. Will a third reviewer be brought in, or will statistical methods be employed to resolve such disagreements?
2/ NIH Study Quality Assessment Tools is appropriate, but consider including a brief explanation of how studies with mixed results or medium-quality assessments will be treated in the meta-analysis.
Data analysis and synthesis
1/ The authors mention using I² and Cochran’s chi-square to assess heterogeneity, but does not discuss strategies to handle studies with significant heterogeneity. Consider outlining how the authors will deal with this—for example, through sensitivity analyses or subgroup analyses.
Discussion
1/ The authors could explore the potential role of psychosocial interventions, such as counseling, in improving QoL post-surgery, as well as the impact of ongoing support groups and rehabilitation programs.
Limitations
limitation of high heterogeneity is noted, but there could be further elaboration on how this will affect the generalizability of the findings. Additionally, clarify whether the exclusion of non-English studies might introduce bias, particularly in regions with high rates of breast cancer but limited English-language publications.
All the best.
Moderate editing of English language needed.
Author Response
Thank you for the review. Kindly find attached "Response_Reviewer_4.docx" file.

Round 2
Reviewer 1 Report
Comments and Suggestions for Authors
None.
Comments on the Quality of English LanguageNone.
Reviewer 3 Report
Comments and Suggestions for Authors
Considering the answer from the authors I consider the manuscript acceptable for publication.